# Assessment of the Seafloor Topography Accuracy in the Emperor Seamount Chain by Ship-Based Water Depth Data and Satellite-Based Gravity Data

**DOI:** 10.3390/s22093189

**Published:** 2022-04-21

**Authors:** Pengpeng Liu, Shuanggen Jin, Ziyin Wu

**Affiliations:** 1School of Remote Sensing and Geomatics Engineering, Nanjing University of Information Science and Technology, Nanjing 210044, China; 20201248057@nuist.edu.cn; 2School of Surveying and Land Information Engineering, Henan Polytechnic University, Jiaozuo 454000, China; 3Shanghai Astronomical Observatory, Chinese Academy of Sciences, Shanghai 200030, China; 4Zhuhai Fudan Innovation Institute, Creative Valley, Hengqin New District, Zhuhai 518057, China; 5Second Institute of Oceanography, Ministry of Natural Resources, Hangzhou 310012, China; zywu@sio.org.cn

**Keywords:** bathymetry, gravity anomaly, Emperor Seamount Chain, remove–restore

## Abstract

The seafloor topography estimation is very important, while the bathymetry data and gravity data are scarce and uneven, which results in large errors in the inversion of the seafloor topography. In this paper, in order to reduce the influence of errors and improve the accuracy of seafloor inversion, the influence of different resolution data on the inversion topography in the Emperor Seamount Chain are investigated by combining ship water depth data and satellite gravity anomaly data released by SIO V29.1. Through the comparison of different resolution models, it is found that the choice of resolution affects the accuracy of the inversion terrain model. An external comparison is presented by using the international high-precision topography data and check points observations. The results show that with the increase in resolution, the fitting residuals obtained by the scale factor are optimized, and the precision of the terrain model is gradually approaching the S&S V19.1 and GEBCO-2020 models, but is better than the ETOPO1 and SRTM 30 models. By external validation using the check points, the standard deviation of the difference was reduced from 58.92 m to 47.01 m, and the correlation between the inverted terrain and the NGDC grid model was increased from 0.9545 to 0.9953. For recovering the Emperor Seamount Chain terrain, the relative error was gradually decreased with the improvement of resolution. The maximum relative error is reduced from 1.09 of 2′ topography to 0.74 of 10″ topography, and the average error is reduced from 0.04 to 0.01 with an improvement by 32.11%. The terrain error between the inverted terrain model and the NGDC grid model is gradually reduced, while the error percentage is increasing by 25.51% and 21.49% in the range of −50 to 50 m and −100 to 100 m, respectively. Furthermore, the sparse area can effectively reduce the terrain standard deviation and improve the terrain correlation by increasing the resolution through the analysis of different density subsets. The error was decreased most significantly in sparse and dense homogeneous regions with increasing resolution.

## 1. Introduction

Seafloor topography is an important basic marine environmental parameter, which not only provides reliable basic data for marine fields, such as marine geology, marine geophysics, marine ecological protection and so on, but also plays an irreplaceable role in marine economy [1] and geodynamics [2,3]. Currently, the most classic and widely used seafloor topographic survey technology is multi-beam bathymetric technology [4], and the high-precision measurement of seabed topography is achieved using the shipborne echo detection technology. Its remarkable advantage is the high detection accuracy, which can quickly obtain the full coverage seafloor topography in the local sea area. However, it has a low efficiency, narrow coverage and high cost. As a result, less than 20% of the global oceans have achieved multi-beam detection, which is far from the urgent needs of ocean detection and management.

While there are a variety of methods to achieve the shallow sea terrain reproduction using multi-beam bathymetry technology, and a combination with machine learning has been carried out, such as bathymetry lidar, satellite-derived bathymetry, and airborne photogrammetry, etc. Using bathymetry lidar combined with image analysis and machine learning, Lukasz et al. [5] evaluated the applicability of airborne lidar bathymetry to determine the structure of topography. For the influence of deposition in the coastal area, Evangelos et al. [6] trained with high-resolution satellite multispectral satellite images and sonar data, and achieved an error precision of 0.5 m with providing effective observational evidence for coastal engineering in monitoring effective coasts. Panagiotis et al. [7] combined machine learning methods to recover the depth using dense point clouds of images and performed the rectified refraction to produce high-precision bathymetry maps, which demonstrated high potential for bathymetry accuracy, as well as texture and orthographic image quality. The above topographic reproduction methods have high precision in shallow sea areas, but cannot be fully applied to deep sea areas.

With the emergent development of satellite altimetry and gravimetry technology [8], it provides a new technical means to establish a global seafloor topography model. The seafloor topography based on satellite altimetry was inverted based on a linear relationship between the gravity anomaly and the seafloor topography. Smith and Sandwell (SAS) and gravity-geologic methods are the most classical. The Smith and Sandwell method was proposed by Smith and Sandwell based on the Parker formula [9] and Watts model [10], and the sea depth was determined based on the linear relationship between the gravity anomaly and the sounding within a certain wavelength. The gravity-geologic method (GGM) was originally used to measure the height of the bedrock under the sediment, which can be used in topography inversion since the internal density of seawater does not change with depth. The gravitational geological method has a fast calculation speed and the seafloor topography can be inverted by using bathymetry data and gravity anomaly for large ocean areas via determining the density difference of the crust.

In recent years, the seafloor topography has made great progress by gravity anomaly inversion. For example, Sandwell et al. [11] used the ocean gravity data to study the plate structure of the seafloor by the altimetry satellites CryoSat-2 and Jason-1. Calmant et al. [12] obtained geoid data through satellite altimetry technology with introduced two-dimensional inversion technology in the frequency domain, and solved the problem of high-order terms by using the iterative idea, which weakened the influence of the bending strength on the inversion error. However, the assumption of the seabed topography was not considered, the plate model was difficult to match with the actual topography, and the calculation of parameter precision is difficult. Hu et al. [13] compared the spatial domain method, frequency domain method, and least square collocation method of seafloor topography inversion, and validated seafloor topography inversion from the vertical gravity gradient anomaly data and the bathymetry data using the frequency domain method. Ramillien et al. [14] constructed the seafloor topography in the spatial domain by combining satellite gravity anomalies, ship-borne gravity anomalies, and ship-borne bathymetry data using the nonlinear least squares method, which took into account the influence of high-order terms on the inversion. Calmant [15] combined the elevation data, shipborne water depth, and gravity anomaly data, and used the least squares in the airspace to propose an inversion model-based method with high-order terms, which significantly improved the uncertainty and calculated the terrain together. Tozer et al. [16] established a high-precision global bathymetric and topographic model with an accuracy of 15″—SRTM15+. Calmant et al. [17] used the multi-source satellite altimetry technology to obtain the ground water surface height data with considering the compensation mechanism in the region, and used the least squares method to predict the sea depth. Yang et al. [18] used the simulated annealing method to invert the seabed topography, and effectively found the global optimal solution, but the parameters in the inversion process were slow. Sui et al. [19] used the gravity anomaly to carry out the seafloor topography inversion experiment in some areas of the North Pacific based on the SAS method, and discussed the influence of nonlinear quadratic and cubic terms on the results. Fan et al. [20,21] used linear regression analysis technology to estimate seabed topography and multiple regression technology to invert seabed topography, and proposed a least squares configuration inversion method to achieve good results with considering the nonlinear term of seabed topography. However, the current accuracy of seafloor topography inversion is still low, which seriously affects its potential applications.

In order to improve the inversion accuracy of complex seafloor topography, this paper intends to assess the seafloor topography accuracy by combining different resolutions’ water depths and gravity data in the area near the Pacific Emperor Seamount (35°~45° N, 166°~174° E). Through the terrain inversion obtained at different resolutions, the error comparison and accuracy evaluation are carried out, and the influence of the resolution on the inversion of the seamounts is analyzed. In Section 2, the study area and data are introduced. Methods and data processing are shown in Section 3. Results and analysis are presented in Section 4, and finally conclusions are given in Section 5.

## 2. Study Area and Data Source

In this paper, the seafloor topography in the northern region of the Pacific Ocean (35°~45° N and 166°~174° E) is inversed and assessed. In order to consider the edge effect, the inversion range is expanded by 1° outward, and then the experimental area is obtained by cropping. The test region is located in the southern region of the Hawaiian-Emperor Seamount Chain with the complex marine environment and the continuous slopes of seafloor topography, including rolling seamounts, sea reefs, and submerged reefs. The seamounts have a large drop of up to 7 km or more, and the terrain height anomaly can be obviously observed, so that the area has a typical representative. It is of practical significance to carry out seafloor topography inversion and accuracy evaluation.

The bathymetry data used in this study are from the National Geophysical Data Center (NGDC), and the early single-beam bathymetry data have obvious measurement errors. Preliminary analysis found that the data may be obtained only a few hundred meters in the bathymetry data of several kilometers, so the error must be eliminated before use. The high-precision GEBCO sounding data are used as a constraint, and 2% of them are used as the detection limit of gross errors. Then, the moving window is used with 10′ × 10′ as the window size, that the moving step length size is 5′, and the error is eliminated by introducing the moving window with three times of the mean square error. A total of 27,717 ship measuring points were obtained, and the inversion considered the edge effect to intercept 1° outward. A total of 18,323 ship measuring points were selected and divided into control points and check points on average. The cyan is the control point, and the red is the check point (Figure 1). Three different subsets were selected in this area, and the number of data points is shown in Table 1. The gravity anomaly data adopts the SIO V29.1 version provided by Smith and Sandwell provided from the Scripps Institution of Oceanography, University of California San Diego, with a resolution of 1′ free gravity anomaly (Figure 1).

## 3. Methods and Data Processing

The relationship between seafloor topography and gravity anomalies is not linear due to the complex and changeable seafloor topography. In order to establish the relationship between seafloor topography and gravity anomalies, the residual water depth and the medium-wave gravity anomaly is approximately linear in the traditional SAS method, and the topographic inversion is carried out through an idealized scale factor. In the frequency domain, based on the robust linear regression analysis method [22], the linear relationship between seafloor topography and gravity anomaly is determined. According to Parker’s formula [9], the relationship is expressed as:(1)$$H\left(K\right)=Z^{-1}\left(K\right)=\left[1/\left(2\pi G\Delta \rho \right)\right]\cdot exp\left(2\pi kd\right)G\left(K\right)\;$$
where G is the gravitational constant of Earth, Δρ is the density difference between lithosphere and seawater, d is the average depth, k is the radial frequency ($\sqrt{k_{x}^{2}+k_{y}^{2}}$), K  is the (kx,ky=1λx,1λy), where $\left(k_{x},k_{y}\right)$ and $\left(\lambda_{x},\lambda_{y}\right)$ are frequency and wavelength in *x* and *y* directions, respectively, $G\left(K\right)$ and $H\left(K\right)$ are the Fourier transform of gravity anomaly (Δg) and seafloor topography $\left(h\right)$, respectively, and $Z\left(k\right)$ is the response function.

In the frequency domain, the “remove-restore” method [23,24] is used to preprocess the gravity anomaly and bathymetry data for inversion in the study area, which are divided into long wave band (>200 km), medium wave band (20~200 km), and short-wave band (<20 km) in the frequency domain. The steps are as follows:(1)The bathymetry data are gridded by cubic spline interpolation, and grids with different resolutions are established simultaneously with the gravity anomaly.(2)The grid bathymetry model data are introduced into the frequency domain, and the long-wave sea depth model is obtained by using Fourier transform through low-pass filtering (200 km).(3)The long-wave sea depth component of the ship measurement point is interpolated according to long-wave sea depth model, and the residual sea depth is obtained by subtracting the sea depth value of the ship measurement point.(4)Fourier transform was performed on the gravity anomaly. After band-pass filtering and downward continuation (20~200 km), the gravity anomaly in the middle wave band was obtained, and the gravity anomaly component of the ship measuring point was obtained by interpolation.(5)The medium wave gravity anomaly and residual sea depth of the same ship survey control point are fitted by linear regression on the horizontal and vertical axes. The fitting result is the scale factor, as shown in Figure 2.

(6)The scale factor is obtained by fitting point positions, and the medium wave seafloor topography is obtained by multiplying it with the medium wave gravity anomaly model.(7)The long-wave band seafloor topography and the medium-wave band seafloor topography are superimposed to obtain the medium- and long-wave sea depth model, while the medium- and long-wave sea depth of the ship survey control point is obtained by interpolation.(8)The difference between the ship survey control point sea depth is calculated to obtain the ship survey control point short-wave sea depth, and the short-wave sea depth model is obtained by gridding.(9)The inversion seafloor topography model is obtained by superposition of medium- and long-wave seafloor model and short-wave seafloor model.

The specific process is shown in Figure 3.

## 4. Results and Analysis

### 4.1. Comparison of Different Resolution Inversion Topographies

Using gravity anomalies and ship survey data [25], through the above inversion process, the seafloor topographic models with different resolutions in the Southern Ocean region of the Pacific Emperor Seamount chain were obtained by changing the resolution size and data processing under the grid settings at different resolutions [26,27,28,29]. The corresponding inversion models are named Model-2′, Model-1′, Model-30″, Model-20″, Model-15″ and Model-10″, respectively. Figure 4 shows the distribution of fitting residuals at different resolutions. With the increasing resolution, the fitting residuals of ship survey points are significantly different, and Model-10″ residual distribution is significantly better than Model-2′, whose residuals are significantly reduced. According to the information provided by the residuals, the rationality of the analysis method and the reliability of the data. Among them, the sum of squares of error (SSE) represents the effect of random errors, the root mean square error (RMSE) represents the deviation between the observed value and the true value, and the mean absolute error (MAE) can better reflect the actual situation of the predicted value error. The sum of squares of the residuals decreases from 2.97 × 10^9^ m to 4.56 × 10^8^ m as the resolution increases from Model-2′ to Model-10″. In addition, RMSE and MAE decreased from 338.93 m and 241.59 m to 132.74 m and 75.05 m, with a decrease of 60.8% and 68.9%, respectively. The above analysis shows that the increasing the resolution of the inverted terrain can significantly reduce the impact of fitting errors, which improves the precision of the inversion model.

Figure 5 shows the seafloor topography inversion models of the Emperor Seamount Chain Southern Ocean Region with different resolutions. The inversion topographies in different resolution inversion models are not significantly different, but have some differences of subtleties in certain terrain areas, such as seamounts, valleys, and rolling areas. The well-known, high-precision terrain models, such as ETOPO1, S&S V19.1, and GEBCO-2020, are compared with our estimated terrain model for accuracy analysis. Table 2 shows statistical analysis on the data of different seafloor topography models. The results show no significant difference between the maximum and minimum values of the inverted terrain models with different resolutions and various terrain models. The standard deviation is used to evaluate the accuracy of the data with the improvement of the inversion resolution, whose standard deviation (STD) of the terrain data is gradually decreased from 925.22 m of the original Model-2′ to 898.24 m of the Model-10″. At the same time, the precision of terrain inversion with different resolutions is higher than that of the ETOPO1 model. With the improvement of resolution, its precision gradually becomes comparable to the high-precision terrain model. For example, Model-15″ and Model-20″ have the same precision as SRTM 30, S&S V19.1, and GEBCO-2020, and Model-10″ has a slightly higher precision.

In order to further analyze the accuracy of the model, the check points without use in the inversion process are used to interpolate each model to obtain the difference between the corresponding sounding data and the check points, and to compare the accuracy of the models [30,31,32]. Table 3 shows the statistical results of the differences at the check points of different terrain models. It can be seen that due to the premature terrain data, the accuracy of the data is poor, and the standard deviation of the terrain data between the checkpoint and the checkpoint of ETOPO1 and SRTM 30 is relatively large. As the resolution increases, the maximum value of the difference between check points and different inverted terrain check points is reduced from 586.05 m to 380.26 m, the minimum is reduced from −388.36 m to −359.29 m, the standard deviation is reduced from 58.92 m to 47.01 m, and the inverted terrain is optimized. Among them, S&S V19.1 has the best data precision and the smallest standard deviation. With the increase in the resolution, the precision of the inverted terrain model is increased by 26.96% when compared to S&S V19.1. At the same time, the inversion precision is comparable to the S&S V19.1 and GEBCO-2020 terrain models.

Figure 6 shows the comparison of inverted terrain data with different resolutions and bathymetric grid terrain models. According to the seafloor topography difference map, as the resolution increases the difference is gradually decreased, and the inversion effect is better. Table 4 shows the statistical differences between terrains with different resolutions and the NGDC grid model. It can be seen that the change of the resolution will affect the accuracy of the inverted terrain; the maximum and minimum of the difference and the average of the difference are inversely proportional to the change of the resolution. As the resolution increases, the difference in data is gradually decreased, and the precision is gradually improved. At the same time, as the resolution increased from Model-2′ to Model-10″, the standard deviation of the difference between the inverted terrain and the NGDC grid is decreased from 275.80 m to 87.93 m, with a decrease of 68.12%. The correlation of the inverted terrain is increased from 0.9545 to 0.9953, with an increase of 4.27%. Figure 7 shows the proportion distribution of the seafloor topography differences. The difference of the inversion topography is gradually decreased by the resolution increases, and the larger difference is optimized and reduced. The error ratio is gradually increased from 62.41% of Model-2′ to 76.45% of Model-10″ for the range of −50~50 m, and from 77.92% of Model-2′ to 90.20% of Model-10″ for the range of −100 to 100 m.

### 4.2. Analysis of Emperor Seamount Chain Terrain

Currently, the Emperor Seamount Chain is a major problem in the seafloor inversion. It contains continuous seamounts; the seamount topography is highly undulate; the topography is undulating; the seamount drop is up to several kilometers; and the bathymetry data is unevenly distributed in the seamount range. The topography complexity has an impact on the accuracy of the inversion of the seafloor topography under certain circumstances. This paper discusses the influence of the topography complexity on the inversion results in the region, whilst eliminating the limitation of the error factors on the inversion, and improving the inversion of the seamount area. The southern area of the Emperor Seamount Chain is studied within the research area (169° E–172° E, 35° N–45° N). The influence of inversion under different resolutions on the local area of seamounts is analyzed, and the check points are interpolated in different resolutions inverted terrain and existing terrain models, which are used to analyze the precision of local areas through check points.

Table 5 shows the differences in check points. The error is the largest near the seamount, indicating that the complexity of the terrain has a greater impact on the accuracy of the seafloor terrain inversion. By changing the inversion terrain resolutions and comparing the interpolation checkpoint data extracted by the grid terrain models with different resolutions, the inversion precision is higher than the ETOPO1 model and the SRTM 30 model. With the increase in resolution, the precision is gradually equal to that of the S&S V19.1 model. The standard deviation of the difference is decreased as the resolution increases, from 66.13 m of Model-2′ to 43.74 m of Model-el-10″, with an increase of 33.86%. The data is gradually optimized. Figure 8 shows the distribution of the topography difference ratio of the Emperor Seamount Chain and the error is gradually decreasing. The error ratio of −50 to 50 m is gradually increased from 54.43% of Model-2′ to 68.31% of Model-10″, which is an increase of 25.51%, and the error ratio of −100 to 100 m is increased from 68.44% of Model-2′ to 83.15% of Model-10″, with an increase of 21.49% over the original.

Figure 9 shows the relative error distribution of the Emperor Seamount Chain seafloor topography extracted from different resolution inversion topography models. It can be seen from the inversion seafloor terrain that the relative error near the seamount chain is relatively large. Because the valley terrain is flat and the terrain complexity is low, the relative error of the sea valley is lower than that of the seamount, and the inversion data has high accuracy. As the height of the seamount increases, the relative error of the seamount increases, and the complexity of the mountain terrain increases. The undulations will cause the inversion accuracy to gradually decrease, indicating that the complexity of the seafloor topography at different levels has different effects on the seafloor topography inversion. The comparison shows that as the resolution increases, the relative error is gradually decreasing by changing the resolution for inversion. The maximum relative error of Model-2′ is 1.09 and the average value is 0.04, which decrease to 0.74 and 0.01 of Model-10″, and its precision is increased by 32.11%. It shows that the terrain grid data of different resolutions have different effects on the precision of terrain inversion. The higher the resolution is, the higher the inversion effect is and the higher the accuracy is.

### 4.3. Analysis of Different Density Regions

The seafloor topography was analyzed in different environments, and the correlation between bathymetry data and seabed topography inversion was discussed [33]. Here random sub-sampling was performed at the bathymetry data points and the regions with different ship survey point densities were selected for secondary analysis. The spatial uniformity of the samples was maintained, as shown in the three regions of Figure 1A–C. The bathymetry data are sparse in area A, dense but uneven in area B, and uniform and dense in area C. Different resolution inversions were performed in different density regions, and the data statistics are shown in Table 6. The inversion topographic data of the three regions A, B, and C have all been optimized. The inversion data in region A have the most significant improvement. The standard deviation has decreased from 837.81 m to 747.24 m, with a decrease of 10.81%, and the correlation coefficient has increased from 0.89 to 0.99. Followed by area B and area C, the standard deviation decreased from 347.90 m and 414.02 m to 342.75 m and 409.93 m, with a decrease of 1.48% and 0.99%, respectively.

In order to further analyze whether the changing resolutions affect the different density regions, an external check is carried out through the check points in each region, and the data are shown in Table 7. From the standard deviation evaluation of data accuracy, it can be seen that the topographic inversion accuracy in the three different density regions is still improved as the resolution increases. The data clearly show that the precision of area A has been improved most significantly. The maximum and minimum values have dropped from 787.98 m and −209.69 m to 474.45 m and −167.77 m, respectively, with the average dropping value from 14.47 m to 11.15 m, and the standard deviation has dropped from 93.14 m to 75.49 m, with an increase of 18.95% on the original basis. The improvement of terrain precision in areas B and C is not as obvious as that in area A, but there is also a good improvement, among which area B has a higher precision improvement than area C. The standard deviations are dropped from 51.24 m and 42.47 m to 48.86 m and 41.24 m, respectively, with an increase of 4.64% and 2.90%.

Figure 10 shows the topographic difference distribution between the inverted topography and the NGDC grid at different resolutions in different density distribution regions. It can be seen that the distribution of terrain difference is mainly concentrated in the range of −100 m–100 m. With the improvement of resolution, the terrain difference changes within this range. The terrain difference in area A is decreased significantly. The topographic difference ratio of −50 m–50 m is gradually increased from 65.30% of Model-2′ to 83.44% of Model-10″, with an increase of 27.78%. The terrain difference ratio of −100 m–100 m is gradually increased from 78.19% of Model-2′ to 92.30% of Model-10″, with an increase of 18.05%. In area B and area C, the proportion of terrain difference between −50 m and 50 m is increased gradually from 72.43% and 60.16% of Model-2′ to 83.54% and 76.80% of Model-10″, with an increase of 15.34% and 27.66%. The terrain difference ratio of −100 m–100 m is gradually increased from 90.80% and 78.20% of Model-2′ to 97.38% and 91.84% of Model-10″, with an increase of 7.25% and 19.43%.

Through the analysis, it is known that the accuracy of the seabed topography inversion in the area of different densities at ship measuring points is improved by improving the resolution. The improvement is most obvious in the sparse area of density distribution, which proves that the proposed method has good applications in a random sampling of input data.

## 5. Conclusions

In this paper, the seafloor topography of the Pacific Emperor Seamount (34°~46° N, 165°~175° E) is inverted by using different resolutions of water depths and gravity data, and the accuracy of the residual and seamount chain topography was separately analyzed. Through external verification, accuracy evaluation, and error analysis, it is concluded that the residual size and terrain model accuracy are optimized as the resolution increases and the error gradually decreases, which is gradually comparable to the high-precision terrain accuracy. The main conclusions are summarized as follows:(1)With different resolutions of both bathymetry data and gravity anomaly, the seafloor topography in the study area is inverted and assessed by combining gravity with bathymetry. The fitting residual is obviously optimized and the precision is improved. The inversion terrain precision is better than that of the ETOPO1 and SRTM 30 terrain by checkpoint verification. With the improvement of resolution, the precision of inversion terrain is gradually close to the S&S V19.1 and GEBCO-2020 terrain. The seafloor terrain model with different resolution grids is established, and the difference between the model and the ship depth survey is analyzed. With the increase of resolution, the terrain difference is decreased significantly, and the correlation of inversion terrain is increased from 0.9545 to 0.9953 with about 4.27%. At the same time, according to the statistics of terrain difference distribution rate, the proportion of terrain difference distribution is increased gradually from 62.41% of the Model-2′ model to 76.45% of the Model-10″ model in the range of−50 to 50 m, and from 77.92% to 90.20% in the range of−100 to 100 m.(2)As a representative of the terrain complexity in the study area, the seamount chain is extracted and analyzed separately. Due to the terrain, the quality of terrain inversion data is poor and the accuracy is low. Improving the resolution of inversion terrain can significantly improve the quality of inversion data, and thus improves the accuracy of inversion terrain. With the increase of the resolution, the standard deviation of terrain interpolation data at the inspection point is decreased from 66.13 m of Model-2′ to 43.74 m of Model-10″ with 33.86%. The regional error and topographic difference of seamounts are decreased, and the proportional distribution of errors in the range of 50 m and 100 m is increased by 25.51% and 21.49%, respectively. From Model-2′ to Model-10″, the maximum relative error and average value are 1.09 and 0.04, which are reduced to 0.74 and 0.01, respectively, and the precision is improved by 32.11%.(3)A subset of ship survey points with different densities are selected for analysis and the inversion accuracy can be more effectively improved by improving the inversion terrain resolution in the sparse area. The standard deviation of topographic inversion data is decreased from 837.81 m to 747.24 m with 10.81% in in area A, while it is decreased by 1.48% and 0.99% in area B and area C, respectively. Comparing with the check points, the standard deviation of area A is decreased from 93.14 m to 75.49 m with 18.95%, while it increased by 4.64% and 2.90% in area B and area C, respectively. For the comparison of terrain differences in different density areas, the proportion of terrain difference between 50 m–50 m in area A is gradually increased from 65.30% of Model-2′ to 83.44% of Model-10″, with an increase of 27.78%. In area B and area C, it is increased by 15.34% and 27.66%, respectively. The improving resolution can also effectively reduce the inversion terrain difference in areas with uniform density. Therefore, the increasing inversion resolution with random sampling of input data can effectively improve inversion data in sparse areas and significantly reduce terrain differences in dense areas.

## Figures and Tables

**Figure 1 sensors-22-03189-f001:**
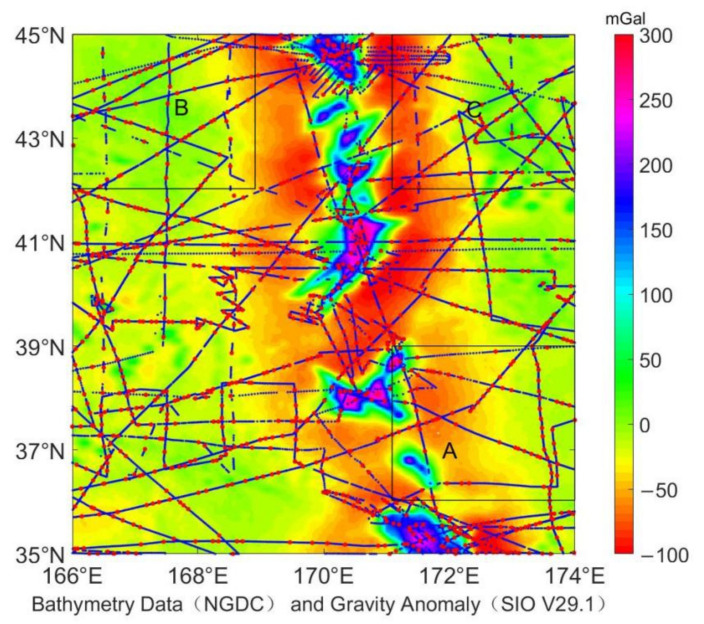
Bathymetry data: control points in cyan and check points in red; gravity anomaly: it comes from SIO V29.1.

**Figure 2 sensors-22-03189-f002:**
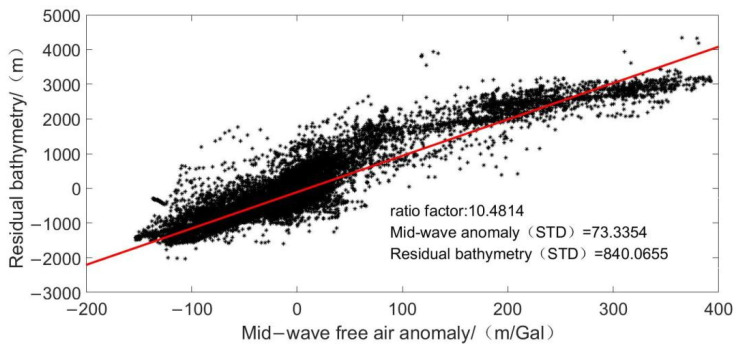
Linear regression results of residual sea depth and gravity anomaly in middle wave section at ship measuring point.

**Figure 3 sensors-22-03189-f003:**
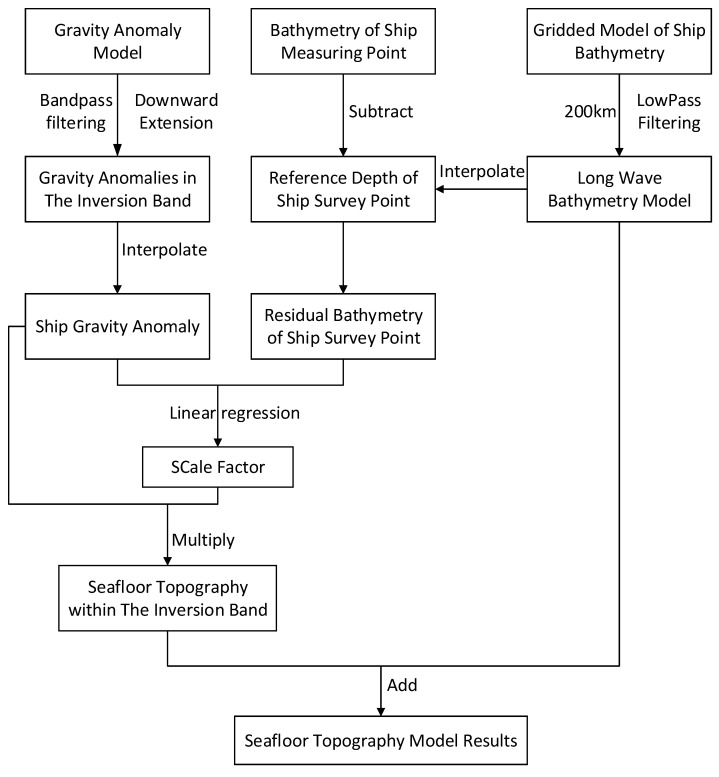
Inversion flow chart of seafloor topography processing.

**Figure 4 sensors-22-03189-f004:**
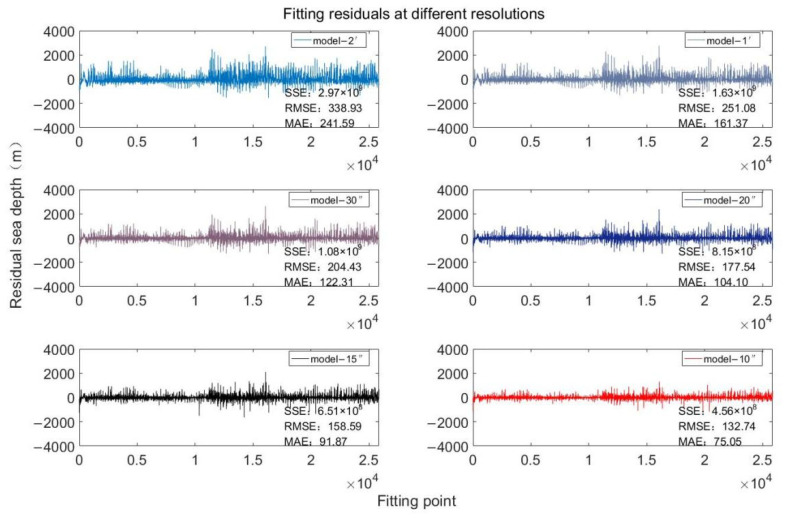
The residual distribution at different resolution inversion topographies.

**Figure 5 sensors-22-03189-f005:**
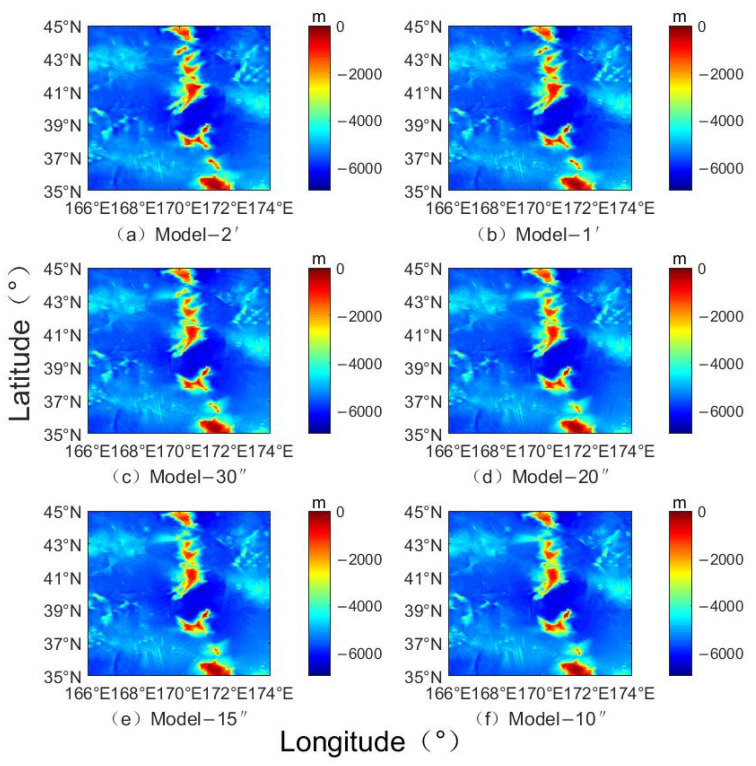
Topographic models with different resolution grid inversions (m).

**Figure 6 sensors-22-03189-f006:**
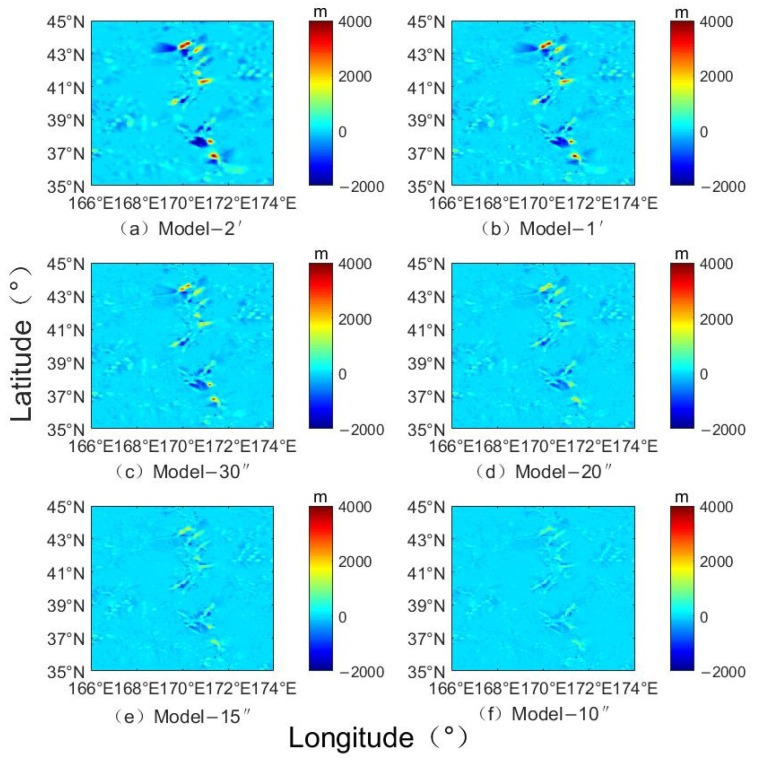
The difference distribution between NGDC grid and inversed topography with different resolutions (m).

**Figure 7 sensors-22-03189-f007:**
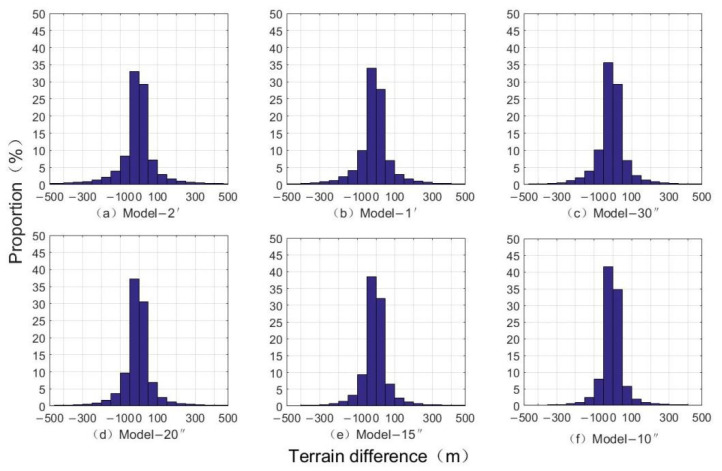
The proportion distribution of inversed seafloor topography difference ratios.

**Figure 8 sensors-22-03189-f008:**
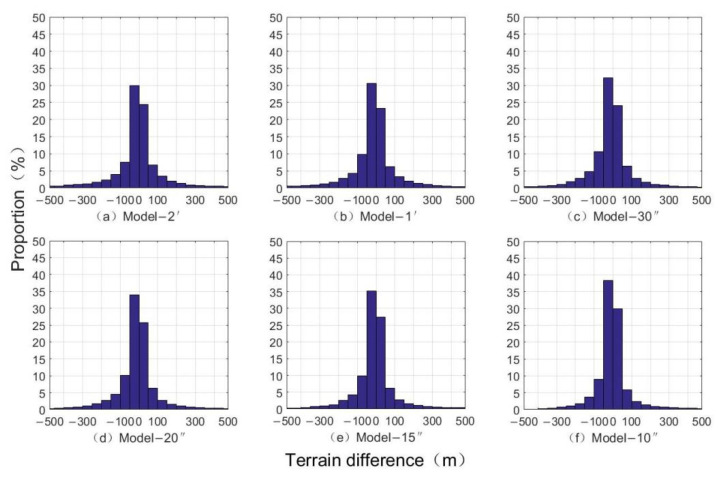
The proportion distribution of topographic differences of the Emperor Seamount Chain.

**Figure 9 sensors-22-03189-f009:**
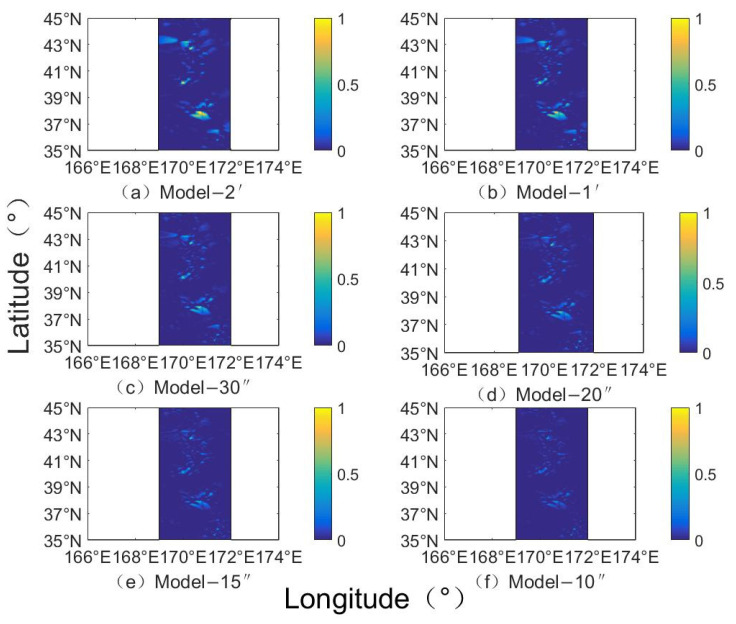
The relative error distribution of the Emperor Seamount Chain seafloor topography between different resolution inversion models and bathymetry grid model.

**Figure 10 sensors-22-03189-f010:**
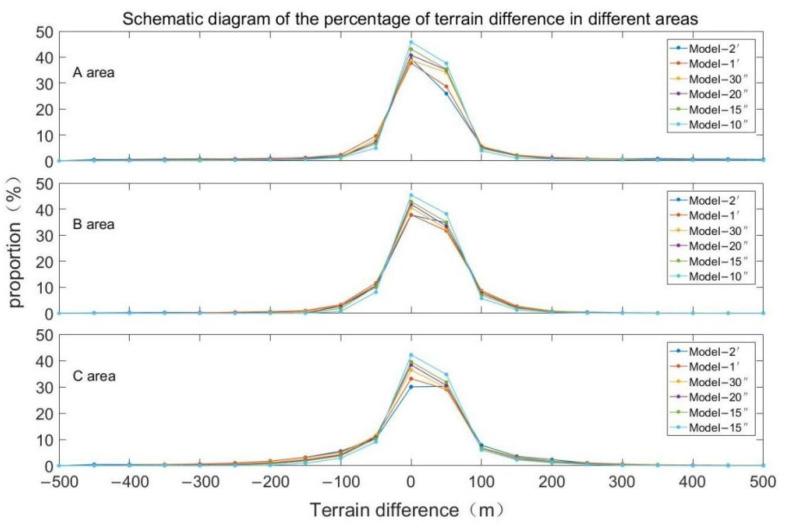
Proportion distribution of topographic differences in retrieved topographical terrain with different resolutions in different density areas.

**Table 1 sensors-22-03189-t001:** Visual statistics of the number of control points and check points in each region.

Number of Points (Pieces)	Total Data Points	Study Area	Area A	Area B	Area C
control points	25,870	16,729	1264	1239	1728
check points	1847	1594	159	155	216

**Table 2 sensors-22-03189-t002:** Statistical analysis of various resolution seafloor topographic models (m).

Topographic Model/(m)	Max	Min	Mean	Rms	Std
Model-2′	−158.32	−6898.42	−5173.33	5255.41	925.22
Model-1′	−81.30	−6611.29	−5173.09	5253.75	917.11
Model-30″	−155.28	−6633.58	−5173.45	5252.90	910.14
Model-20″	−189.80	−6597.58	−5174.02	5252.66	905.51
Model-15″	−190.33	−6587.99	−5174.53	5252.61	902.29
Model-10″	−213.53	−6624.66	−5175.27	5252.65	898.24
ETOPO1	31.00	−6646.00	−5178.10	5262.57	939.11
SRTM30	−2.00	−6743.00	−5193.40	5275.21	905.42
S&S V19.1	−3.33	−6715.76	−5189.40	5272.67	903.36
GEBCO-2020	−2.29	−6752.55	−5189.45	5273.04	905.17

**Table 3 sensors-22-03189-t003:** Difference statistics of topographic model at check point (m).

Difference between Model and Check Point/(m)	Max	Min	Mean	Std
Model-2′	586.05	−388.36	0.94	58.92
Model-1′	541.27	−372.04	0.82	49.90
Model-30″	493.24	−376.75	0.73	48.46
Model-20″	458.44	−375.94	0.69	47.98
Model-15″	428.54	−370.07	0.52	47.57
Model-10″	380.26	−359.29	0.46	47.01
ETOPO1	1489.23	−882.20	7.27	121.84
SRTM30	968.53	−506.95	5.65	78.02
S&S V19.1	425.33	−290.08	4.07	44.18
GEBCO-2020	430.13	−360.23	4.30	48.18

**Table 4 sensors-22-03189-t004:** The difference between terrain and NGDC grid topography at different resolutions (m).

Model Differences/(m)	Max	Min	Mean	Std	Correlation
Model-2′	4026.32	−2493.2	3.11	275.80	0.9545
Model-1′	3972.49	−2210.59	3.29	237.23	0.9660
Model-30″	3166.21	−1804.01	2.93	184.55	0.9793
Model-20″	2542.13	−1525.24	2.36	148.97	0.9864
Model-15″	2083.64	−1245.02	1.85	123.15	0.9907
Model-10″	1388.06	−895.12	1.11	87.93	0.9953
ETOPO1	2866.07	−2154.66	−4.50	210.13	0.9608
strm30	2253.77	−2312.18	−4.95	198.15	0.9768
S&S V19.1	1953.73	−2014.64	−2.31	113.76	0.9863
GEBCO	2073.51	−1798.10	−2.92	112.13	0.9859

**Table 5 sensors-22-03189-t005:** Difference data statistics at check points of various topographic models of the Emperor Seamount Chain (m).

Difference between Model and Check Point/(m)	Max	Min	Mean	Std
Model-2′	586.05	−388.36	1.78	66.13
Model-1′	541.27	−372.04	1.51	55.39
Model-30″	493.24	−376.75	1.71	52.32
Model-20″	458.44	−375.94	1.53	48.54
Model-15″	428.54	−370.07	1.45	46.62
Model-10″	380.26	−359.29	1.55	43.74
ETOPO1	1489.23	−882.2	−11.04	234.77
SRTM30	968.53	−506.95	−16.87	168.63
S&S V19.1	425.33	−290.08	−4.71	39.10
GEBCO-2020	430.13	−360.23	−3.35	33.46

**Table 6 sensors-22-03189-t006:** Statistics of terrain model data with different resolutions in different density areas (m).

Different Density Areas	Model	Max	Min	Mean	Rms	Std	Correlation
Area A	Model-2′	−377.30	−6230.24	−5284.85	5337.41	837.81	0.8914
	Model-1′	−236.78	−6277.32	−5279.25	5333.83	808.70	0.9206
	Model-30″	−215.21	−6231.03	−5255.69	5330.96	783.69	0.9514
	Model-20″	−189.80	−6302.90	−5274.97	5329.49	770.62	0.9694
	Model-15″	−155.28	−6295.51	−5270.04	5327.99	761.13	0.9798
	Model-10″	−81.30	−6184.99	−5267.78	5322.04	747.24	0.9900
Area B	Model-2′	−4110.95	−5957.47	−5252.37	5263.71	347.90	0.9736
	Model-1′	−4071.29	−5963.66	−5251.39	5262.56	347.44	0.9769
	Model-30″	−4063.20	−5969.03	−5248.69	5260.14	346.87	0.9845
	Model-20″	−4046.02	−5965.22	−5248.37	5259.89	346.40	0.9879
	Model-15″	−4028.14	−5965.28	−5248.35	5259.84	345.43	0.9902
	Model-10″	−4003.98	−5972.27	−5248.07	5259.49	342.75	0.9935
Area C	Model-2′	−3323.22	−6662.10	−5537.23	5552.68	414.02	0.9653
	Model-1′	−3281.33	−6534.84	−5532.70	5547.88	411.44	0.9667
	Model-30″	−3259.03	−6541.64	−5530.44	5545.61	411.24	0.9755
	Model-20″	−3242.02	−6520.40	−5527.96	5543.21	410.84	0.9810
	Model-15″	−3235.82	−6506.26	−5526.18	5541.46	410.08	0.9851
	Model-10″	−3161.68	−6492.97	−5523.95	5539.25	409.93	0.9909

**Table 7 sensors-22-03189-t007:** Statistics of difference data of terrain model check points in different density areas (m).

Different Density Areas	Model	Max	Min	Mean	Std
Area A	Model-2′	787.98	−209.69	14.47	93.14
	Model-1′	541.27	−174.77	14.87	89.13
	Model-30″	493.24	−174.10	13.67	81.74
	Model-20″	462.11	−173.70	12.52	80.14
	Model-15″	456.73	−171.43	11.82	77.16
	Model-10″	474.45	−167.77	11.15	75.49
Area B	Model-2′	277.21	−191.97	−3.11	51.24
	Model-1′	240.97	−191.37	−2.81	50.72
	Model-30″	236.91	−188.90	−2.70	50.44
	Model-20″	232.13	−182.06	−2.29	49.96
	Model-15″	236.39	−168.07	−1.41	49.43
	Model-10″	227.40	−153.90	−1.04	48.86
Area C	Model-2′	199.90	−165.25	8.54	42.47
	Model-1′	151.40	−154.10	8.50	42.03
	Model-30″	156.06	−147.37	8.47	41.87
	Model-20″	157.28	−143.69	8.05	41.44
	Model-15″	156.23	−139.46	7.87	41.33
	Model-10″	153.55	−130.00	7.36	41.24

## Data Availability

Gravity anomaly data are downloaded from https://topex.ucsd.edu/cgi-bin/get_data.cgi (last accessed: 24 February 2022). and Bathymetry data are downloaded from https://www.ncei.noaa.gov/maps/iho_dcdb (last accessed: 26 February 2022).

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
