# Peer review of "Assessment of the Seafloor Topography Accuracy in the Emperor Seamount Chain by Ship-Based Water Depth Data and Satellite-Based Gravity Data"

_sensors, 2022, doi:10.3390/s22093189_

Round 1
Reviewer 1 Report
Please find the file attached

Author Response
Responses:
Thanks very much for your comments and good suggestion. In the revised manuscript, we fully agreed your suggestion and have added these analyses following your suggestion.
Please see the attachment

Reviewer 2 Report
see attached

Author Response

(The authors gave the same response as above.)

Round 2
Reviewer 2 Report
see attached

Author Response
Please see the attachment

This manuscript is a resubmission of an earlier submission. The following is a list of the peer review reports and author responses from that submission.
Round 1
Reviewer 1 Report
In this paper, water depths and gravity data of different resolutions were taken to inverse the seafloor topography in the Emperor Seamount Chain area. With the shipborne gravimetry data as external check, the accuracy was analyzed.
However, the method used in this paper is not so novel which means lack of innovation.
Many conceptual errors exist in the description, e.g. Equation (1) is a expression of frequency domain, not spatial domain. Figure 3, the flow chart is confused with many mistakes.
In the study area, 1847 points shipborne data were used as external check. from 27717 points data. The proportion is significantly lower than the proportion normally used. So the credibility of the results is questionable.
Besides, the English writing needs to be greatly enhanced.
Author Response
Hello teacher,I have answered your questions one by one,please see the attachment.

Reviewer 2 Report
The manuscript is devoted to the topic of seafloor topography which is of importance and relevance. Authors, in order to reduce the influence of errors and improve the accuracy of seafloor inversion, aim to analyze the influence of different resolution data on the inversion topography by combining bathymetry data and gravity anomaly data. Authors must presents further information and the details of the proposed external comparison which involved the international high-precision topography data and check points observations The validation needs elaboration specially on the involvement of the check points and standard deviation. Several statements and equations need correct reference and citations.State of the art needs improvement. A detailed description of the cited references is essential. Several recently published papers are not included in the review section. In fact, the acknowledgment of the past related work by others, in the reference list, is not sufficient. Consequently, the contribution of the paper is not clear. Furthermore, consider elaborating on the suitability of the paper and relevance to the journal. Kindly note that references cited must be up to date.
Elaborate on the method used and why used this method.
Limitations and validation are not discussed adequately. The research question and hypothesis must be answered and discussed clearly in the discussion and conclusions. Please communicate the future research. The lessons learned must be further elaborated in the conclusion by discussing the results to the community and the future impacts. What is your perspective on future research? how may machine learning could be used to improve the modeling? for instance the proposed models in "Groundwater quality assessment for sustainable drinking and irrigation" or "Ensemble Boosting and Bagging Based Machine Learning Models for Groundwater Potential Prediction" please elaborate.
The originality of the paper is not discussed well. The research question must be clearly given in the introduction, in addition to some words on the testable hypothesis. Please elaborate on the importance of this work. Please discuss if the paper suitable for broad international interest and applications or better suited for the local application? Elaborate and discuss this in the introduction.
Explains acronymus and abbreviation when first appear. Insert a acronymus table as well.
Author Response

(The authors gave the same response as above.)

Round 2
Reviewer 1 Report
- The method is lack of innovation.
- The credibility of the results is questionable.
- The manuscipt writing needs to be improved.
These problems which existed in last version still not amended and solved in this version.
Reviewer 2 Report
The comments addressed, just some referencing and citations have errors due to Endnote error.
